# Dementia Literacy in the Greater Bay Area, China: Identifying the At-Risk Population and the Preferred Types of Mass Media for Receiving Dementia Information

**DOI:** 10.3390/ijerph17072511

**Published:** 2020-04-07

**Authors:** Angela Y. M. Leung, Alex Molassiotis, June Zhang, Renli Deng, Ming Liu, Iat Kio Van, Cindy Siu U Leong, Isaac S. H. Leung, Doris Y. P. Leung, Xiaoling Lin, Alice Y. Loke

**Affiliations:** 1School of Nursing, The Hong Kong Polytechnic University, Hong Kong 999077, China; alex.molasiotis@polyu.edu.hk (A.M.); isaacleung201709@gmail.com (I.S.H.L.); doris.yp.leung@polyu.edu.hk (D.Y.P.L.); alice.yuen.loke@polyu.edu.hk (A.Y.L.); 2World Health Organization Collaborating Centre for Community Health Services, The Hong Kong Polytechnic University, Hong Kong 999077, China; 3School of Nursing, Sun Yat-Sen University, Guangzhou 510080 China; zhangje@mail.sysu.edu.cn (J.Z.); linxling3@mail.sysu.edu.cn (X.L.); 4Department of Nursing, The 5th Affiliated Hospital of Zhuyi Medical University, Zhuhai 519100, China; renli.deng@gmail.com; 5School of Health Sciences and Sports, Macao Polytechnic Institute, Macau 999078, China; karryliu@ipm.edu.mo (M.L.); suleong@ipm.edu.mo (C.S.U.L.); 6Kiang Wu Nursing College of Macau, Macau 999078, China; van@kwnc.edu.mo; 7Department of Statistics, The Chinese University of Hong Kong, Hong Kong 999077, China

**Keywords:** health literacy, dementia, cross-sectional study, community, Alzheimer’s Disease, knowledge

## Abstract

*Background:* The aim of this cross-sectional study was to assess the dementia literacy of community-dwelling adults in four cities (Hong Kong, Guangzhou, Macau, and Zhuhai) of the Greater Bay Area of China, and to determine their mass media preferences for receiving dementia information. *Methods:* The survey was completed by 787 community-dwelling adults. Dementia literacy was indirectly measured using two validated scales—the 30-item Alzheimer’s Disease Knowledge Scale and the 20-item Dementia Attitude Scale (DAS). Participants were also asked to indicate whether they wanted to receive dementia information via digital or traditional media. Chi-square tests, logistic regressions, and MANOVA analyses were conducted. *Results:* Unemployed or retired people had poor attitudes towards dementia and lower levels of knowledge about dementia. Single, cohabiting, or divorced people in Hong Kong and Macau had lower DAS scores than married people. Young people and those with a secondary education preferred to get their dementia information from social media. People with a tertiary education and employed people enjoyed searching government or hospital websites for information. Middle-aged, unemployed, or retired people tended to learn about dementia from television or radio. *Conclusion:* It is worth educating the public about dementia and developing strategies consistent with their preferences for types of mass media.

## 1. Introduction

Dementia is a growing health concern in many countries, including China. In 2016, there were 46.8 million people in the world living with dementia, about 25% of them in China [1]. A recent meta-analysis of 96 studies showed that the pooled prevalence of dementia was 5.3% in mainland China and 7.2% in Hong Kong [2]. As the number of people suffering from dementia increased, more attention was paid to the dementia literacy of the public. The Greater Bay Area is a newly developed integrated economic and business zone in China, and is home to 69 million people. It consists of seven cities (including Guangzhou and Zhuhai) and two special administrative regions (Hong Kong and Macau). In line with the growth of economic activity in the Greater Bay Area, the local governments have been spent substantial sums on healthcare to keep their citizens healthy. Both Hong Kong and Macau have their own healthcare systems, which differ from those of Guangzhou and Zhuhai. At US$319 billion, Hong Kong’s gross domestic product (GDP) was the highest, its dependency rate was the lowest (22.2%), and its dementia prevalence was the highest in the region (see the Appendix A; Table A1). There were an estimated 9.55 million persons with dementia in China and Hong Kong [2]. As the population of this area aged, dementia has received a lot of attention in the last decade. Dementia literacy has become one of the most discussed issues in the Greater Bay Area.

Dementia literacy refers to knowledge of dementia and attitudes towards persons with dementia [3]. With inadequate dementia literacy, individuals might not have enough knowledge to recognize the symptoms of dementia, identify treatment options, and provide essential care and support to persons with dementia [4]. In addition, individuals with inadequate dementia literacy might feel a tremendous amount of discomfort when persons with dementia are around them, or would not feel obliged to support these people and their caregivers [4]. Social discomfort with people with dementia has been found to be prevalent, but this phenomenon could be reversed by increasing knowledge about dementia by educating the public [5]. To date, most studies in dementia literacy were from high-income countries, such as the study conducted in Australia by Low and team [3,6] and that conducted by Woo [7] in the USA. There are few such studies from low- and middle-income countries [8]. China is one of those countries where the prevalence of dementia is growing rapidly; therefore, dementia literacy has become an important topic for discussion. The project team has identified three studies [9,10,11] on dementia literacy in China. A survey with 140 participants in the Chinese city of Tianjin showed that only 16% of the participants knew the risk factors of dementia and 56% were not sure whether they should share the diagnosis of dementia with the patients [9]. Another study indicated that the majority of the respondents (77%) had a personal fear of developing Alzheimer’s Disease (AD), and that females, those with poor self-rated health, those in close proximity to someone with AD, and those with a high perception of the severity of AD, were more likely to indicate such a fear [10]. A recent study investigated the dementia literacy of older adults (those aged 60 or above) living in 34 urban cities in China and found that their dementia literacy was very low [11]. The studies mentioned above had investigated the Chinese participants’ basic knowledge of dementia, their intended actions if their own relatives were to suffer from dementia, their personal fear of developing dementia, and the dementia literacy level of older adults. However, it was not clear from past research in China what would be the characteristics of those at risk for inadequate dementia literacy. This study intended to provide evidence of dementia literacy among community-dwelling adults (aged 18–64) and to investigate which specific groups of community-dwelling adults had low dementia literacy.

Mass media play an important role in health promotions. There is evidence that the mass media promote healthy behaviors (such as safe sex and the prevention of skin cancer) through descriptive norms [12]. Nonetheless, the role of the mass media in dementia education has seldom been investigated. Would traditional mass media still be the best choice for promoting dementia literacy? Would digital media become the favored channels for receiving information about dementia? Knowing the preferred channels would help in the effort to develop appropriate strategies for disseminating relevant health information to the public. The aim of this study was to assess the dementia literacy level of community-dwelling adults in four cities (Hong Kong, Guangzhou, Macau, and Zhuhai) of the Greater Bay Area of China, and their preferences in types of mass media for receiving information on dementia. Older adults were excluded from this study because Zhang and team [11] had already investigated the dementia literacy of older adults.

## 2. Materials and Methods

### 2.1. Design, Data Collection, and Participants

This is a multi-city, cross-sectional study with the samples representing their respective ratio of the population in the selected cities/regions. The target was to recruit 200 subjects in each city. A convenience sample was used in this study. The study population consisted of adults aged 18 or above, who were cognitively intact (Short Portable Mental Status Questionnaire, SPMSQ > 7), able to read Chinese, and had lived in their respective city for at least 1 year. Those who were hospitalized or living in a residential care home were excluded. In this study, we only included respondents aged 18–64 years, in the analysis. University students in different cities were trained to be data collectors. The number of data collectors varied in each city—5 students each in Macau and Guangzhou, 10 in Hong Kong, and 27 in Zhuhai. Before they embarked on the process of collecting data, all of data collectors underwent a 2-day training session held by the project team. The collectors approached as many people as they could in community centers or public places (such as bus-stops, parks, shopping malls, and markets), screened them for eligibility to participate in this study, obtained their written consent to participate, and provided assistance to those who needed help in completing the questionnaires. Data collection was conducted from July 2017 to May 2018, in the four cities.

### 2.2. Measures

The outcome measure was dementia literacy. Since there was no single validated instrument for measuring dementia literacy, we measured the variable indirectly by mapping the two most relevant concepts—knowledge about dementia and attitudes towards dementia, as suggested by Low and Anstey [3].

Knowledge about dementia was measured using the Alzheimer’s Disease Knowledge Scale (ADKS). The ADKS contains 30 true/false questions that assess knowledge related to AD assessment, diagnosis, caregiving, life impact, prevalence, prevention, risk factors, symptoms, treatment, and management [13]. A composite score was calculated by counting the number of questions answered correctly, and the score could range from 0 to 30. The ADKS was shown to have good internal consistency (alpha = 0.71) and adequate test-retest reliability (reliability coefficient = 0.81) [13].

Attitudes towards dementia were measured using the Dementia Attitudes Scale (DAS), which is a 20-item instrument. Each item is rated on a 7-point Likert scale ranging from 1 (strongly disagree) to 7 (strongly agree), and the total score of the scale could be from 20 to 140. The higher the total score, the more positive was the attitude toward dementia. The result of the confirmatory factor analysis indicated that DAS is a model with a reasonably good fit. The Cronbach’s alpha of this scale was 0.83 [14].

Both ADKS and DAS contributed to dementia literacy and were considered dependent variables in the study.

Another outcome measure was the channels for receiving dementia information. The participants were asked to indicate their preferred types of mass media for receiving relevant information about dementia, and they could choose more than one type. A total of 12 options were given. The options were then categorized into four types—social media, websites, paper-based materials, and television/radio. Social media included WhatsApp, Facebook, QQ, Instagram, Baidu, WeChat, and blogs, while websites referred to websites from government units, hospitals, health-related organizations, and non-governmental organizations. Paper-based materials referred to posters, newspapers, magazines, and books.

### 2.3. Ethical Statement

Written consent was obtained from the eligible subjects before the completion of the questionnaires. Ethical approval to conduct the study was obtained from the Human Subject Ethics Subcommittee of Sun-Yat Sen University, the Hong Kong Polytechnic University (HSEARS20170511002), Kiang Wu Hospital College of Nursing of Macau, and Zhuyi Medical University of Zhuhai.

### 2.4. Statistical Analysis

Descriptive statistics were used to summarize the sample characteristics. Considering that both ADKS and DAS contribute to dementia literacy, we put two continuous variables into the multivariate analysis of variance (MANOVA) model as the dependent variables. Two-way MANOVA was used because two or more independent variables could be tested simultaneously with two dependent variables. Demographics (gender, education level, employment level, and accommodation) and location (Hong Kong, Macau, Zhuhai, and Guangzhou) were the independent variables. Interaction terms between each demographic factor and location were also put into the model to assess the effect of demographics on dementia literacy, in different cities.

MANOVA was used after validating the normality and homogeneity assumptions. The existence of multivariate outliers was assessed by a test of normality, and outliers were eliminated, when detected. The normality of the data was also assessed using the Shapiro–Wilk test; normality was assumed at *p* > 0.05. We then assessed the Mahalanobis distance of the two dependent variables (ADKS and DAS). Since the Mahalanobis distance of this data was 13.44 (that is, less than the critical value of the Chi-square of 13.82), multivariate normality was assumed. According to Levene’s test (*p* > 0.05), the criterion group variances were homogenous. We assessed the maximum likelihood criteria (Wilk’s lambda), and effect size was presented as partial eta (η^2^). When the MANOVA detected significant statistical differences between the IVs and the DVs, we proceeded to conduct a two-way ANOVA for the dependent variables and a post-hoc test, to assess the effect of categorical independent variables on the dependent variable.

Four logistic regression models, using a Wald stepwise backward elimination method with the probability of a stepwise entry alpha of 0.05 and removal alpha of 0.10, were conducted to identify the associated factors (demographics, the cities where the participants lived, dementia knowledge, and attitudes) in favor of a particular type of mass media, for receiving information on dementia. All statistical analyses were performed using the Statistical Package for the Social Sciences (SPSS 25.0, IBM Corp., Armonk, NY, USA). The level of significance was set at *p* < 0.05.

## 3. Results

### 3.1. Demographic Characteristics and Level of Dementia Literacy

A total of 787 community-dwelling adults (aged 64 or below) were included in the analysis, with 242 from Hong Kong (HK), 184 from Guangzhou (GZ), 177 from Macau (MA), and 184 from Zhuhai (ZH). Table 1 shows the demographic characteristics of the participants. The majority of the participants were employed. About half of the participants in Hong Kong, Guangzhou, and Macau were married, but significantly more participants in Zhuhai were married. In terms of level of education, Hong Kong had the highest percentage of participants possessing a university-level degree among the four cities/regions. About half of the participants were living in private housing in Hong Kong, Guangzhou, and Zhuhai.

### 3.2. Profiles of Participants with Inadequate Dementia Literacy

Table 2 shows the results of the MANOVA with ADKS and DAS as the dependent variables. Of all demographic factors, only employment status (students, employed, unemployed) had a significant effect on dementia literacy, Wilk’s Λ = 0.984, F(4, 1492) = 3.001, *p* < 0.05, and partial η^2^ = 0.008. This indicated that students, the employed, and the unemployed had significantly different levels of dementia literacy. Looking at the individual components of dementia literacy (ADKS and DAS), employment status had a significant effect on DAS (F(2, 747) = 4.130, *p* < 0.05, and partial η^2^ = 0.011), but not on ADKS (see the Appendix A; Table A2). This meant that employment affected people’s attitudes towards dementia, but not knowledge about dementia. Unemployed or retired people (mean = 82.01, SD = 12.28) had a lower DAS score than employed people (mean = 84.49, SD = 11.38) and students (mean = 88.73, SD = 12.17) (see the Appendix A; Table A3). However, the effect of employment status on dementia literacy was different in four cites, with Wilk’s Λ = 0.969, F(12, 1492) = 1.971, *p* < 0.05, and partial η^2^ = 0.016. Figure 1 shows that unemployed or retired people had the lowest dementia knowledge in all cities, while students in Hong Kong had the highest ADKS scores among all people.

Age did not affect dementia literacy as a whole, but its effect was significantly different in the four cities, with Wilk’s Λ = 0.980, F(6, 1492) = 2.510, *p* < 0.05, and partial η^2^ = 0.010 (Table 2). In addition, there was a significant difference in ADKS when age was considered, with F(1, 747) = 3.861, *p* < 0.05, and partial η^2^ = 0.005 (see the Appendix A; Table A2). This implies that age affected knowledge about dementia but did not influence people’s attitudes towards dementia.

Although marital status did not have any effect on dementia literacy, its effect on DAS was significant. The patterns in the four cities were different, with F(3, 747) = 3.494, *p* < 0.05, and partial η^2^ = 0.014 (see the Appendix A; Table A2). Figure 2 shows that single, cohabiting, or divorced people in Hong Kong and Macau had lower DAS scores than married people, but the opposite was observed in Zhuhai and Guangzhou.

### 3.3. Preferences in Mass Media

Figure 3 shows the preferences for types of mass media for receiving dementia information, in the four cites. Among the four types of media, there was no significant difference in website preference among the participants of all four cites. Table 3 shows the characteristics of those participants who preferred digital media. Social media were more likely to be preferred by those who had a secondary education (OR 1.584) and were less likely to be preferred by those who were older (OR 0.950). Participants from Guangzhou (OR 3.713), Macau (OR 1.851), and Zhuhai (OR 3.153) were more likely to use social media to receive information on dementia than the Hong Kong participants. On the other hand, those participants who were students (as reference) and employed people (OR 1.303) were more likely and participants with only a primary education (OR 0.454) were less likely to prefer to get dementia information from government or hospital websites (Table 3).

We also investigated the preferences with regard to traditional media for health promotions—paper-based materials, television, or radio. Older participants (OR 1.013, *p* < 0.05) preferred to get dementia-related information from these traditional media (Table 4). Hong Kong (as reference) and Macau (OR 1.733) participants were more likely to enjoy reading paper-based materials to receive dementia-related information. On the other hand, unemployed/retired people (OR 1.320) and students (as reference) preferred to watch TV or listen to the radio, to learn about dementia. Compared to the Hong Kong participants, Guangzhou (OR 0.473) and Macau (OR 0.518) people were less likely to select television or radio as their preferred medium for receiving dementia-related information.

## 4. Discussion

This was the first study to assess the level of dementia literacy in the Greater Bay Area of China. Its findings echo those of the studies conducted in other countries, such as Australia [6], Cambodia, Fiji, and the Philippines [15], showing that dementia literacy was inadequate in our sample. This study filled in a missing piece of the puzzle with regard to dementia literacy in China by providing further evidence of inadequate dementia literacy among community-dwelling adults (aged 18–64 years) in addition to old adults (aged 65 years or above) [11]. Public education to improve citizens’ knowledge of dementia (including an understanding of its symptoms, treatment options, care needs, and prognoses) and to cultivate positive attitudes towards dementia should be more widely supported. The Hong Kong Special Administrative Region Government has already made educating the public about dementia a part of its mental health policy, highlighting the strategy of bringing together education, housing, police, healthcare, social care authorities, and research institutions to co-create dementia-friendly communities [16]. For example, in education settings, school curricula are to be modified to include lifestyle modifications with regards to dementia. Training would be given to the police force so that policemen will have a better understanding of the symptoms and the behaviors of persons with dementia. When necessary, police force will search for persons with dementia who might have gotten lost or who might have wandered away from home. These initiatives are good examples to refer to when increasing city-based, awareness-raising efforts with the participation of people from different sectors. Although some of these initiatives have already been instituted, the actual impacts of these initiatives on the dementia literacy of Hong Kong residents have not been investigated in the current study. Further studies are warranted to assess how these initiatives have affected Hong Kong residents’ understanding of dementia symptoms and their acceptance of persons with dementia in the neighborhood.

We identified vulnerable populations with inadequate dementia literacy. These findings could help to focus resources on improving the dementia literacy of citizens in individual cities/regions. Unemployed or retired people seem to be a vulnerable group with a low dementia literacy. Digital media are their preferred channel to learn about dementia. Health service providers and government agents should consider using digital media to provide learning opportunities to this specific group. Other than building up knowledge of dementia symptoms, diagnoses, and treatment options, attention should be given to cultivating positive attitudes towards dementia. Short videos (3–5 min) are a useful tool for changing attitudes. Discussions about attitudes can be made in the chatrooms of social media sites.

Our finding that married people in Hong Kong and Macau had more positive attitudes towards dementia than their single counterparts, contributed to our understanding of the relationship between marital status and dementia literacy. Most previous studies had found that married people had more knowledge about dementia than unmarried people [17,18]. However, our finding showed another perspective—that of attitudes towards dementia. As dementia literacy is a complex concept that includes both knowledge and attitudes, this aspect of attitudes should not be neglected. In response to this finding, dementia attitude training could be offered to unmarried people in Hong Kong and Macau. It is worth noting that attitudes appear to be different in Zhuhai and Guangzhou, where married people were found to have a less positive attitude than unmarried people. A further study should be conducted to determine why married people in Zhuhai and Guangzhou hold such an attitude, and how we could rectify this situation.

As noted, Hong Kong and Macau residents and middle-aged people prefer paper-based materials for dementia education. Materials should be developed by taking into consideration the literacy levels of the readers. The use of pictures or photographs in educational materials is recommended. Comic books have been found to be effective at improving knowledge about medication adherence, the storage of drugs, and health literacy, among Chinese people [15]. Fotonovelas (or comic books illustrated with photos) have been shown to increase the overall knowledge about dementia among older Latino adults, whose level of education is generally not high [19]. With this evidence, we could consider developing photo-based interventions or using comic books to promote dementia literacy.

Using social media to deliver public education in dementia was a strategy favored by the participants in the current study, particularly those from the young generation. Such a proposal was worth supporting, as social media are already embedded in daily life and would be a novel medium for disseminating public health information [20]. Nonetheless, to date, few empirical studies have been carried out to promote dementia using social media. One of the two related studies identified was a qualitative study involving 42 adolescents aged 12–18 in the United Kingdom, who discussed the key features of effective dementia education [21]. These adolescents proposed using Facebook, Instagram, and YouTube to facilitate their learning about dementia [21]. The findings of the current study support this recommendation of using social media to educate the public about dementia, but our findings imply that the target could be extended to middle-aged adults. Another pilot study explored the use of YouTube to deliver dementia education to older adults aged 55 or above, and found that more participants (aged 65 or above) had used YouTube to learn about dementia in the second year than in the first year of the study [22]. Thus, the potential was huge for different types of social media to deliver quality dementia education to the public. WeChat, Baidu, and QQ are the frequently used social media platforms in Guangzhou and Zhuhai, while WhatsApp, Facebook, and YouTube are common in Hong Kong and Macau; these were the platforms that were chosen by the participants in this study. A further investigation is warranted to determine how these social media platforms could be used in dementia education in the Chinese population. In Hong Kong, a pilot test of the use of Facebook to improve the dementia literacy of healthcare professionals was carried out [23]. It was found to be feasible and was welcomed by the participants [23]. There is a possibility that such work would be extended to the general Chinese population. Regardless of which types of social media are used, ethical issues such as the protection of privacy and confidentiality should be considered [24].

Improving dementia literacy requires the use of multiple channels and techniques [7]. Traditional mass media such as television and radio were the preferred channels in Hong Kong and Zhuhai to get information about dementia. Through these channels, a large audience could be reached in a short period of time. Nonetheless, there have been no study on the impacts of television and radio episodes on changes in dementia literacy. Future research in this particular area is warranted.

Dementia literacy is a multi-faceted concept, and its definition has been inconsistent in many previous studies [25]. In the current literature, dementia literacy often refers to knowledge and beliefs about dementia, and the ability to access, understand, and utilize health information in dementia care, and prevention. In this study, we adopted Low and Anstey’s [3] definition of dementia literacy and used two validated scales, ADKS and DAS, to measure dementia literacy. This was the first attempt to combine these two concepts as one and to use MANOVA to perform the analysis. This method of statistical analysis allows us to review the combined effect of knowledge and attitudes on dementia literacy, and the effect of each separate concept. In future research, other related concepts (such as financial literacy and caregiving literacy) should also be investigated.

This study had several limitations. First, the use of a cross-sectional design limited the extent to which causal inferences of the observed associations could be established. The findings should be further examined in longitudinal studies to establish the temporal validity of any associations that were found. Second, some of the important factors, such as personal experience in dementia care or the impact of Chinese culture on dementia, were not measured. This precluded us from examining their independent and potential interaction effects on dementia literacy and preferences in the types of media for receiving dementia information. This could also partly explain the limited explanatory power of the associated factors for the outcome variables. Finally, the current sample was obtained using convenience sampling, which might not have produced representative samples of the population in the four cities. This limits the generalizability of the current findings to the larger population of the four cities.

## 5. Conclusions

Since unemployed and retired people in the Greater Bay Area of China have inadequate dementia literacy, it is worth conducting public education about dementia, targeting these groups of citizens. Strategies targeted at specific groups of citizens in each city could be formulated according to their preferences in different types of mass media. These recommendations are in response to the World Health Organization’s Global Calls for Action, to develop and implement campaigns to raise public awareness about dementia and foster dementia-inclusive societies by 2025 [26].

## Figures and Tables

**Figure 1 ijerph-17-02511-f001:**
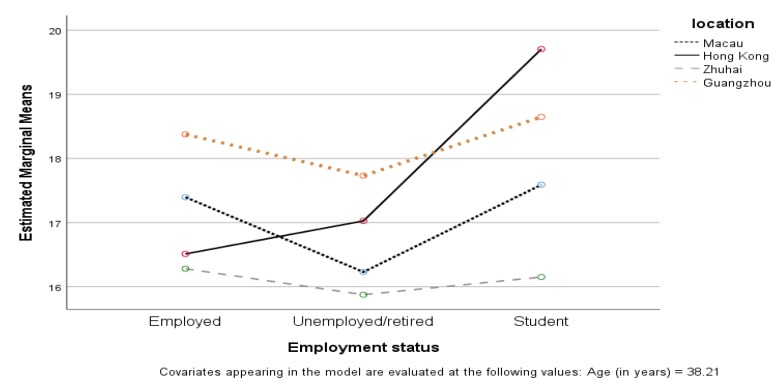
Comparison of Alzheimer’s Disease Knowledge Scale (ADKS) scores among students, and the employed and unemployed participants across four cities.

**Figure 2 ijerph-17-02511-f002:**
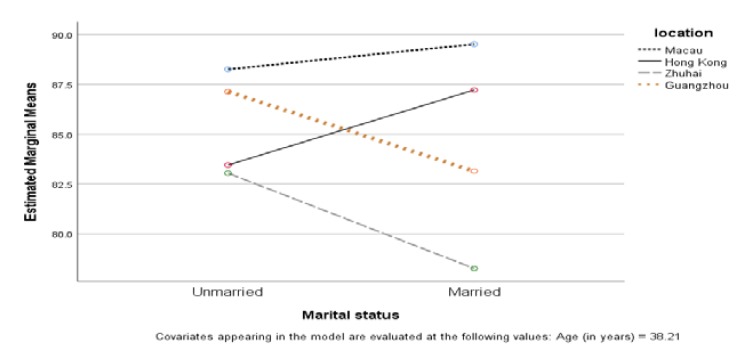
Comparison of Dementia Attitude Scale (DAS) scores among married people and others across four cities.

**Figure 3 ijerph-17-02511-f003:**
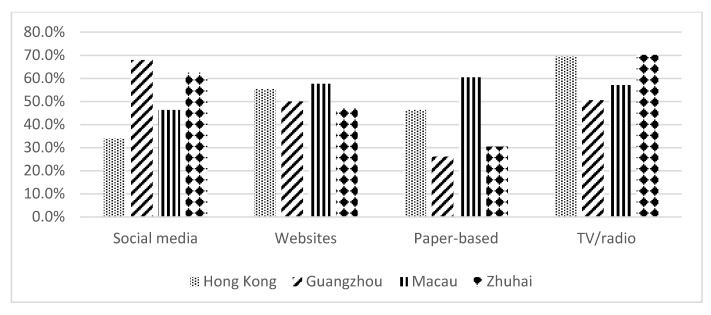
Preferences in mass media for receiving dementia information.

**Table 1 ijerph-17-02511-t001:** Demographic data of the participants by city.

	Hong Kong		Guangzhou		Macau		Zhuhai	
	Count	%	Count	%	Count	%	Count	%
Age (mean, SD)	38.86	14.94	34.08	13.11	41.02	14.05	38.80	13.32
0–44	128	52.9%	126	68.5%	110	62.1%	106	57.6%
45–64	114	47.1%	58	31.5%	67	37.9%	78	42.4%
Gender								
Female	136	56.2%	93	50.5%	96	54.2%	92	50.0%
Male	106	43.8%	91	49.5%	81	45.8%	92	50.0%
Education								
≤Primary level	24	9.9%	9	4.9%	22	12.4%	27	14.7%
Secondary level	94	38.8%	92	50.0%	85	48.1%	134	72.8%
Tertiary level	124	51.2%	83	45.1%	70	39.5%	23	12.5%
Marital status								
Singled/Cohabiting/Divorced	128	52.9%	90	48.9%	82	46.3%	67	36.4%
Married	114	47.1%	94	51.1%	95	53.7%	117	63.6%
Employment status								
Employed	167	69.0%	104	56.5%	120	67.8%	126	68.5%
Unemployed	35	14.5%	26	14.1%	46	26.0%	40	21.7%
Student	40	16.5%	54	29.3%	11	6.2%	18	9.8%
Accommodation								
Private	117	48.3%	95	51.6%	123	69.5%	104	56.5%
Public	92	38.0%	20	10.9%	20	11.3%	4	2.2%
Others	33	13.6%	69	37.5%	34	19.2%	76	41.3%

**Table 2 ijerph-17-02511-t002:** Associations between demographic factors and dementia literacy using multivariate analysis of variance (MANOVA).

Effect	Wilks’ Lambda	F	Df	Error df	*p* Value	Partial Eta Squared
Intercept	0.267	1025.63	2	746	<0.000	0.733
Main effect						
Gender	0.995	1.979	2	746	0.139	0.005
Employment status	0.984	3.001	4	1492	0.018	0.008
Accommodation	0.996	0.759	4	1492	0.552	0.002
Education	0.992	1.421	4	1492	0.225	0.004
Marital status	0.997	1.259	2	746	0.284	0.003
Age	0.995	1.971	2	746	0.140	0.005
Interaction term						
Gender × location	0.996	0.466	6	1492	0.833	0.002
Employment × location	0.969	1.971	12	1492	0.023	0.016
Accommodation × location	0.981	1.181	12	1492	0.291	0.009
Education × location	0.982	1.158	12	1492	0.309	0.009
Marital status × location	0.985	1.911	6	1492	0.076	0.008
Age × location	0.980	2.510	6	1492	0.020	0.010

**Table 3 ijerph-17-02511-t003:** Preferred digital media for receiving dementia information.

	Social Media		
	Odds Ratio	95% CI	*p*
Age (in years)	0.950	0.936, 0.965	<0.001
Employment status			
Students	Ref		
Employed	1.639	1.002, 2.681	0.049
Unemployed/Retired	1.235	0.633, 2.407	0.536
Education			
Tertiary	Ref		
Secondary	1.584	1.074, 2.336	0.020
Primary	0.895	0.463, 1.730	0.742
Location			
Hong Kong	Ref		
Guangzhou	3.713	2.413, 5.715	<0.001
Macau	1.851	1.208, 2.834	0.005
Zhuhai	3.153	2.013, 4.940	<0.001
Constant	2.018		0.012
Nagelkerke R^2^		0.219	
	**Websites**		
Education			
Tertiary	Ref		
Secondary	1.018	0.733, 1.413	0.915
Primary	0.454	0.263, 0.784	0.005
Accommodation			
Others	Ref		
Private	1.074	0.765, 1.508	0.682
Public	1.591	1.017, 2.488	0.042
Employment status			
Students	Ref.		
Employed	1.303	0.856, 1.983	0.217
Unemployed/Retired	0.825	0.478, 1.426	0.491
ADKS	1.044	0.999, 1.091	0.058
Constant	0.440		0.080
Nagelkerke R^2^		0.043	

Note. Social media included Facebook, QQ, Instagram, Baidu, WeChat, and blogs, while websites referred to websites from government units, hospitals, health-related organizations, and non-governmental organizations.

**Table 4 ijerph-17-02511-t004:** Preferred traditional media for receiving dementia information.

	Paper-Based Materials		
	Odds Ratio	95% CI	*p*
Age	1.013	1.003, 1.024	0.015
Cities			
Hong Kong	Ref		
Guangzhou	0.434	0.285, 0.659	<0.001
Macau	1.733	1.167, 2.573	0.006
Zhuhai	0.506	0.338, 0.758	0.001
Constant	0.517		0.008
Nagelkerke R^2^		0.103	
	**Television/Radio**		
Age	1.021	1.007, 1.034	0.002
Employment			
Students	Ref		
Employed	0.750	0.469, 1.200	0.230
Unemployed/retired	1.320	0.681, 2.556	0.411
DAS	1.013	1.000, 1.027	0.056
Cities			
Hong Kong	Ref		
Guangzhou	0.473	0.314, 0.712	<0.001
Macau	0.518	0.341, 0.787	0.002
Zhuhai	1.096	0.711, 1.690	0.677
Constant	0.403		0.179
Nagelkerke R^2^		0.084	

Note. Paper-based materials refer to posters, newspapers, magazines, and books.

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
