# Peer review of "Dementia Literacy in the Greater Bay Area, China: Identifying the At-Risk Population and the Preferred Types of Mass Media for Receiving Dementia Information"

_ijerph, 2020, doi:10.3390/ijerph17072511_

Round 1

Reviewer 1 Report

The authors investigated dementia literacy in 4 cities in China, risk factors and preferred media for receiving information. Since dementia literacy is required for early recognition of dementia symptoms and seeking medical advice and for providing adequate support to patients, such studies are always welcomed.

The study is well designed and conducted and the statistics are adequate.

Only one point: Do the authors have an explanation for the effect of marital status on dementia literacy (why married persons in Zhuhai were more likely to have inadequate literacy)?

Author Response

Response 1: As suggested by another reviewer, we performed two-way MANOVA and consider both ADKS and DAS as the dependent variables simultaneously. So the above finding (married people in Zhuhai were more likely to have inadequate literacy) was removed from the text.

In the MANOVA results, we found that marital status significantly affected attitudes (DAS), but not dementia knowledge (ADKS). The following sentences are added to Discussion (page 10)

Our finding that married people in Hong Kong and Macau had more positive attitudes towards dementia than their single counterparts contributed to our understanding of the relationship between marital status and dementia literacy. Most previous studies had found that married people had more dementia knowledge than unmarried people [17,18]. But our finding showed another perspective – that of attitudes towards dementia. As dementia literacy is a complex concept that includes both knowledge and attitudes, the aspect of attitudes should not be neglected. In response to this finding, dementia attitude training could be offered to unmarried people in Hong Kong and Macau. It is worth noting that attitudes appear to be different in Zhuhai and Guangzhou, where married people were found to have a less positive attitude than unmarried people. A further study should be conducted to determine why married people in Zhuhai and Guangzhou hold such an attitude, and how we could rectify this situation.

Reference:

  1. Van Patten, R., & Tremont, G. (2018). Public knowledge of late-life cognitive decline and dementia in an international sample. Dementia (London), 1471301218805923. doi: 10.1177/1471301218805923.

  1. Tan WJ, Hong SI, Luo N, Lo TJ, Yap P. (2012). The lay public’s understanding and perception of dementia in a developed Asian Nation. Dementia and Geriatric Cognitive Disorder Extra, 2(1), 433-44. doi: 10.1159/000343079.

Reviewer 2 Report

Interesting study. To assess the magnitude of the human effort involved in collecting information, this reviewer is left with the lack of information of how many data collectors (university students) contributed to the study in each city. This information is not available in the manuscript,  but it could be helpful to provide an idea of this aspect of the study to other investigators contemplating similar epidemiologic approaches.  

Author Response

Response to Reviewer 2 Comments

Point 1: Interesting study. To assess the magnitude of the human effort involved in collecting information, this reviewer is left with the lack of information of how many data collectors (university students) contributed to the study in each city. This information is not available in the manuscript,  but it could be helpful to provide an idea of this aspect of the study to other investigators contemplating similar epidemiologic approaches.  

Response 1: The following sentence was added on page 3

The number of data collectors varied in each city: 5 students each in Macau and Guangzhou, 10 in Hong Kong, and 27 in Zhuhai. Before they embarked on the process of collecting data, all of the data collectors underwent a 2-day training session held by the project team.

Reviewer 3 Report

Comments and Suggestions for Authors

This paper covers an important knowledge gap about dementia literacy in the Greater Bay Area of China, particularly in younger and middle-aged adults, which had not been previously explored. Although this is an important topic that we need to understand more, I have serious reservations regarding their statistical methods and overall quality of writing.

  1. Arbitrary cut-off scores
    The most important critique of this paper is arbitrarily determined cutoff scores for the ADKS and DAS without citing external studies. Furthermore, converting continuous variables into categorical variables is an approach that should be avoided unless there are clear reasons to do so (please see Royston et al., 2006, Statist, Med). Dichotomizing a continuous variable loses information and reduces statistical power. Especially for this study, because there is not an established clinical cut-off score for the ADKS and DAS, we would not know that the cut-off scores represented what the authors intended to represent. Please re-analyze the data using the linear regression to keep the ADKS and DAS as continuous variables.
  1. A rationale for 4 separate models for each city
    Another concern regarding the authors’ statistical approach is the rational for running 4 separate regression models for each city, instead of running one model with combining all the cities and including the city as a covariate (for Results 3.2). Testing one model with whole sample will have more statistical power and also increase an interpretability. Please display this main result of the study in a graph/table. 
  1. Visual representation of the data
    As a related note above, the authors should judiciously select a small number of effective visual representations for their data throughout the paper. For instance, for the media preferences in each city displayed in Table 2, visual representations could be used (such as a bar graph).
  1. Please add more description for selection criteria of backward elimination.
  2. Please explain how the internal consistency was calculated. The method section did not mention that the same participants took the survey more than once. How was the test-retest score calculated?

  3. Several items that need more specifications

6-a. The introduction nicely described the relative lack of research on dementia literacy in Eastern and lower-income countries. However, in the introduction, the sentence, “To the best knowledge of the project team, three studies in relation to dementia literacy were identified.” does not specify whether these studies were conducted in China, low-income countries, or just around the world. This should be specified, and also these studies should be cited. Additionally, if these three studies on dementia literacy are indeed in China, it should be clarified what this study adds that the others do not cover.

6-b. The last sentences of the introduction clearly state the purpose and aims of this study, but these aims should also clarify why only adults 18-64 years of age were recruited and not older adults as well. A study of dementia literacy in older adults in China was mentioned in the previous paragraph (Zhang et al., 2017), but it should be restated here in the aims that this study covers an age range that study did not.

6-c. In line 152, please be more specific about “the associate factors of participants.”

6-d. In the first paragraph of the results section, it states, “A total of 844 people completed the survey. Among these, 788 community-dwelling adults (aged 64 or below) were included in the analysis…” without explaining why the other 56 were not included. Please clarify this discrepancy.

  1. An interpretation of housing type in Discussion
    The paper states that housing and poverty are connected, however, and that per capita annual income has been measured in previous papers Leung et al. (2019) and Jackson et al. (2009) (but not this one). Thus, housing seems to be an indirect measure of poverty, which may be the actual relevant variable. I am not convinced that examining housing is truly helpful, especially since income/poverty was not examined directly in this study.
  1. Potential limitations or future directions regarding education initiatives by Hong Kong Sar Government
    Suggestions to improve dementia literacy based on media preference and age were discussed in detail. Public education initiatives in particular were mentioned. However, it was not clear whether the dementia public education initiatives the Hong Kong Sar Government has as part of its mental health policy have already been instituted or are proposed future measures. If they have already been instituted, are there any data to show whether these measures have been effective? If not, collecting these data should be suggested for future studies.

  1. Dementia literary as multi-faceted concept
    The authors mentioned that there is no single validated instrument to measure dementia literacy. In addition, dementia literacy is also multi-faceted concept (Choi, Rose, Friedman, 2018, Gerontro Geriatr Med). Please include this point in the discussion.

  1. We highly recommend that the authors work with English editors to polish the quality of writing. We identified several spelling and grammatical errors throughout, including misspelling Alzheimer’s Disease in the abstract as “Alzhiemer’s Disease” (line 33), inadequate as “inadquate” (line 180), marital as “marrital” (line 184), and dementia as “demential” (line 189).

Author Response

Response to Reviewer 3 Comments

This paper covers an important knowledge gap about dementia literacy in the Greater Bay Area of China, particularly in younger and middle-aged adults, which had not been previously explored. Although this is an important topic that we need to understand more, I have serious reservations regarding their statistical methods and overall quality of writing.

Point 1: Arbitrary cut-off scores
The most important critique of this paper is arbitrarily determined cutoff scores for the ADKS and DAS without citing external studies. Furthermore, converting continuous variables into categorical variables is an approach that should be avoided unless there are clear reasons to do so (please see Royston et al., 2006, Statist, Med). Dichotomizing a continuous variable loses information and reduces statistical power. Especially for this study, because there is not an established clinical cut-off score for the ADKS and DAS, we would not know that the cut-off scores represented what the authors intended to represent. Please re-analyze the data using the linear regression to keep the ADKS and DAS as continuous variables.

Response 1: We did the analyses again with two-way Multivariate Analysis of Variance (MANOVA). The following sentences were added to the Method Section

Page 3: Both ADKS and DAS contributed to dementia literacy and were considered dependent variables in the study.

Page 4: Considering that both ADKS and DAS contribute to dementia literacy, we put the two continuous variables into the Multivariate Analysis of Variance (MANOVA) model as the dependent variables. Two-way MANOVA was used because two or more independent variables could be tested simultaneously with two dependent variables. Demographics (gender, education level, employment level, and accommodation) and location (Hong Kong, Macau, Zhuhai, and Guangzhou) were the independent variables. Interaction terms between each demographic factor and location were also put into the model to assess the effect of demographics on dementia literacy in different cities.

MANOVA was used after validating the normality and homogeneity assumptions. The existence of multivariate outliers was assessed by a test of normality, and outliers were eliminated, when detected. The normality of the data was also assessed using the Shapiro-Wilk test; with p > 0.05, normality was assumed. We then assessed the Mahalanobis distance of the two dependent variables (ADKS and DAS). Since the Mahalanobis distance of this data is 13.44 (that is, less than the critical value of the Chi-square of 13.82), multivariate normality was assumed. According to Levene’s test (p > 0.05), the criterion group variances were homogenous. We assessed the maximum likelihood criteria (Wilk’s lambda). Effect size was presented as Partial eta (η2). When the MANOVA detected significant statistical differences between the IDVs and DVs, we proceeded to conduct a two-way ANOVA for the dependent variables and a post-hoc test to assess the effect of categorical independent variables on the dependent variable.

Point 2: A rationale for 4 separate models for each city
Another concern regarding the authors’ statistical approach is the rational for running 4 separate regression models for each city, instead of running one model with combining all the cities and including the city as a covariate (for Results 3.2). Testing one model with whole sample will have more statistical power and also increase an interpretability. Please display this main result of the study in a graph/table. 

Response 2: We use a new variable ‘location’ to indicate the four cities. The whole sample is now included. The main MANOVA result is shown in Table 2, and the following sentences were added to Results Section (page 6):

Table 2 shows the results of the MANOVA with ADKS and DAS as the dependent variables. Of all the demographic factors, only employment status (students, employed, unemployed) had a significant effect on dementia literacy, Wilk's Λ = 0.984, F(4, 1492) = 3.001, p < 0.05, partial η2 = 0.008. This indicated that students, the employed, and the unemployed had significantly different levels of dementia literacy. Looking at the individual components of dementia literacy (ADKS and DAS), employment status had a significant effect on DAS (F(2, 747) = 4.130, p < 0.05, partial η2 = 0.011), but not on ADKS (see the supplementary material: Table A2). That means that employment affects people’s attitudes towards dementia, but not dementia knowledge. Unemployed or retired people (mean, SD = 82.01, 12.28) had a lower DAS score than employed people (mean, SD 84.49, 11.38) and students (mean, SD = 88.73, SD 12.17) (see the supplementary material: Table A3). However, the effect of employment status on dementia literacy was different in the four cites, with Wilk's Λ = 0.969, F(12, 1492) = 1.971, p < 0.05, and partial η2 = 0.016. Figure 1 shows that unemployed or retired people had the lowest dementia knowledge in all cities, while students in Hong Kong had the highest ADKS scores among all people.

Age did not affect dementia literacy as a whole, but its effect was significantly different in the four cities, with Wilk's Λ = 0.980, F(6, 1492) = 2.510, p < 0.05, and partial η2 = 0.010 (Table 2). In addition, there was a significant difference in ADKS when age was considered, with F(1, 747) = 3.861, p < 0.05, and partial η2 = 0.005 (see the supplementary material: Table A2). This implies that age affects dementia knowledge but does not influence people’s attitudes towards dementia.

Although marital status did not have any effect on dementia literacy, its effect on DAS was significant. The patterns in the four cities were different, with F(3, 747) = 3.494, p < 0.05, and partial η2 = 0.014 (see the supplementary material: Table A2). Figure 2 shows that single, cohabiting, or divorced people in Hong Kong and Macau had lower DAS scores than married people, but the opposite was observed in Zhuhai and Guangzhou.

The following sentences were added to Discussion (page 10):

Unemployed or retired people seem to be a vulnerable group with low dementia literacy. Digital media are their preferred channel to learn about dementia. Health service providers and the government agents should consider using digital media to provide learning opportunities to this specific group. Other than building up knowledge of dementia symptoms, diagnoses, and treatment options, attention should be given to cultivating positive attitudes towards dementia. Short videos (3-5 minutes) are a useful tool for changing attitudes. Discussions about attitudes can be made in the Chatroom of social media sites.

Our finding that married people in Hong Kong and Macau had more positive attitudes towards dementia than their single counterparts contributed to our understanding of the relationship between marital status and dementia literacy. Most previous studies had found that married people had more dementia knowledge than unmarried people [17,18]. But our finding showed another perspective – that of attitudes towards dementia. As dementia literacy is a complex concept that includes both knowledge and attitudes, the aspect of attitudes should not be neglected. In response to this finding, dementia attitude training could be offered to unmarried people in Hong Kong and Macau. It is worth noting that attitudes appear to be different in Zhuhai and Guangzhou, where married people were found to have a less positive attitude than unmarried people. A further study should be conducted to determine why married people in Zhuhai and Guangzhou hold such an attitude, and how we could rectify this situation.

Point 3: Visual representation of the data
As a related note above, the authors should judiciously select a small number of effective visual representations for their data throughout the paper. For instance, for the media preferences in each city displayed in Table 2, visual representations could be used (such as a bar graph).

Response 3: Page 8: A bar graph is now shown in Figure 3 which replace the original Table 2.

Point 4: Please add more description for selection criteria of backward elimination.

Response 4: The following sentence was added to the Method section (page 4):

Four logistic regression models, using a Wald stepwise backward elimination method with the probability of a stepwise entry alpha of 0.05 and removal alpha of 0.10, were conducted to…

Point 5: Please explain how the internal consistency was calculated. The method section did not mention that the same participants took the survey more than once. How was the test-retest score calculated?

Response 5: This is a cross-sectional study with validated scales. Participants completed the survey once, so there was no test-retest score for this survey.

Point 6: Several items that need more specifications

6-a. The introduction nicely described the relative lack of research on dementia literacy in Eastern and lower-income countries. However, in the introduction, the sentence, “To the best knowledge of the project team, three studies in relation to dementia literacy were identified.” does not specify whether these studies were conducted in China, low-income countries, or just around the world. This should be specified, and also these studies should be cited. Additionally, if these three studies on dementia literacy are indeed in China, it should be clarified what this study adds that the others do not cover.

Response 6-a: These three studies were conducted in China, and we have highlighted the key findings of these studies in the subsequence sentences. We modify the sentence and added the reference number in the sentence (page 2):

The project team identified three studies [9, 10, 11] on dementia literacy in China. A survey with 140 participants in the Chinese city of Tianjin showed that only 16% of the participants knew the risk factors of dementia and 56% were not sure whether they should share the diagnosis of dementia with the patients [9]. Another study indicated that the majority of the respondents (77%) had a personal fear of developing Alzheimer’s Disease (AD), and that females, those with poor self-rated health, those in close proximity to someone with AD, and those with a high perception of the severity of AD were more likely to indicate such a fear [10]. A recent study investigated the dementia literacy of older adults (those aged 60 or above) living in 34 urban cities in China and found that their dementia literacy was very low [11]…. This study intends to provide evidence of dementia literacy among community-dwelling adults (aged 18-64) and find out which specific groups of community-dwelling adults have low dementia literacy.

6-b. The last sentences of the introduction clearly state the purpose and aims of this study, but these aims should also clarify why only adults 18-64 years of age were recruited and not older adults as well. A study of dementia literacy in older adults in China was mentioned in the previous paragraph (Zhang et al., 2017), but it should be restated here in the aims that this study covers an age range that study did not.

Response 6-b:  The following sentence was added (page 3):

Older adults were excluded from this study because Zhang and team [11] had already investigated the dementia literacy of older adults.

6-c. In line 152, please be more specific about “the associate factors of participants.”

Response 6-c: The sentence was modified to (page 4):

identify the associated factors (demographics, the cities where the participants lived, dementia knowledge and attitudes) in the favoring of a particular type of mass media…

6-d. In the first paragraph of the results section, it states, “A total of 844 people completed the survey. Among these, 788 community-dwelling adults (aged 64 or below) were included in the analysis…” without explaining why the other 56 were not included. Please clarify this discrepancy.

Response 6-d: Before conducting MANOVA, we removed the outliers. One subject was considered as the outlier. Therefore there was a minor change in the sample size (from 788 to 787). To avoid confusion to the audience, we removed the first sentence and reported the actual number of samples included in the analysis. Those 56 subjects were the people aged 65 or above, so we did not include them in the analysis. The following sentence was modified (page 4):

A total of 844 people completed the survey. Among these, 787 community-dwelling adults (aged 64 or below) were included in the analysis, with 242 from Hong Kong (HK), 184 from Guangzhou (GZ), 177 from Macau (MA) and 184 from Zhuhai (ZH).

Point 7: An interpretation of housing type in Discussion
The paper states that housing and poverty are connected, however, and that per capita annual income has been measured in previous papers Leung et al. (2019) and Jackson et al. (2009) (but not this one). Thus, housing seems to be an indirect measure of poverty, which may be the actual relevant variable. I am not convinced that examining housing is truly helpful, especially since income/poverty was not examined directly in this study.

Response 7: Since the MANOVA results did not indicate the relationship between accommodation and dementia literacy, the above relevant sentences were deleted.

Point 8: Potential limitations or future directions regarding education initiatives by Hong Kong Sar Government
Suggestions to improve dementia literacy based on media preference and age were discussed in detail. Public education initiatives in particular were mentioned. However, it was not clear whether the dementia public education initiatives the Hong Kong Sar Government has as part of its mental health policy have already been instituted or are proposed future measures. If they have already been instituted, are there any data to show whether these measures have been effective? If not, collecting these data should be suggested for future studies.

Response 8: The following sentence was added in Discussion (page 10):

Although some of these initiatives have already been instituted, the actual impacts of these initiatives on the dementia literacy of Hong Kong residents have not been investigated in the current study. Further studies are warranted to assess how these initiatives have affected Hong Kong residents’ understanding of dementia symptoms and their acceptance of persons with dementia in the neighborhood.

Point 9: Dementia literary as multi-faceted concept
The authors mentioned that there is no single validated instrument to measure dementia literacy. In addition, dementia literacy is also multi-faceted concept (Choi, Rose, Friedman, 2018, Gerontro Geriatr Med). Please include this point in the discussion.

 Response 9: The following sentences were added (page 11):

Dementia literacy is a multi-faceted concept, and its definition has been inconsistent in many previous studies [25]. In the current literature, dementia literacy often refers to knowledge and beliefs about dementia, the ability to access, understand, and utilize health information in dementia care, and prevention. In this study, we adopted Low and Anstey’s [3] definition of dementia literacy and used two validated scales, ADKS and DAS, to measure dementia literacy. This is the first attempt to combine these two concepts as one and to use MANOVA to perform the analysis. This method of statistical analysis allows us to review the combined effect of knowledge and attitudes on dementia literacy, and the effect of each separate concept. In future research, other related concepts (such as financial literacy and caregiving literacy) should also be investigated.

Point 10: We highly recommend that the authors work with English editors to polish the quality of writing. We identified several spelling and grammatical errors throughout, including misspelling Alzheimer’s Disease in the abstract as “Alzhiemer’s Disease” (line 33), inadequate as “inadquate” (line 180), marital as “marrital” (line 184), and dementia as “demential” (line 189).

Response 10: This version was sent to professional editing.

Round 2

Reviewer 3 Report

Authors addressed all the concerns that I raised. I appreciate that they incorporated necessary statistical tests into the study very carefully. The manuscript is now well-written and conveys important scientific results effectively.

This manuscript is a resubmission of an earlier submission. The following is a list of the peer review reports and author responses from that submission.